# Chrysin Is Immunomodulatory and Anti-Inflammatory against Complete Freund’s Adjuvant-Induced Arthritis in a Pre-Clinical Rodent Model

**DOI:** 10.3390/pharmaceutics15041225

**Published:** 2023-04-12

**Authors:** Muhammad Asif Faheem, Tasleem Akhtar, Nadia Naseem, Usman Aftab, Muhammad Shoaib Zafar, Safdar Hussain, Muhammad Shahzad, Glenda Carolyn Gobe

**Affiliations:** 1Department of Pharmacology, University of Health Sciences, Lahore 54600, Pakistan; 2Department of Morbid Anatomy and Histopathology, University of Health Sciences, Lahore 54600, Pakistan; 3Centre for Applied Molecular Biology (CAMB), University of the Punjab, Lahore 53700, Pakistan; 4School of Biomedical Sciences, Faculty of Medicine, University of Queensland, Brisbane, QLD 4072, Australia

**Keywords:** anti-inflammatory, immunomodulatory, bioactive components, chrysin, phytotherapy, rheumatoid arthritis

## Abstract

Chrysin (5,7-dihydroxyflavone) has many pharmacological properties including anti-inflammatory actions. The objective of this study was to evaluate the anti-arthritic activity of chrysin and to compare its effect with the non-steroidal anti-inflammatory agent, piroxicam, against complete Freund’s adjuvant (CFA)-induced arthritis in a pre-clinical model in rats. Rheumatoid arthritis was induced by injecting CFA intra-dermally in the sub-plantar region of the left hind paw of rats. Chrysin (50 and 100 mg/kg) and piroxicam (10 mg/kg) were given to rats with established arthritis. The model of arthritis was characterized using an index of arthritis, with hematological, biological, molecular, and histopathological parameters. Treatment with chrysin significantly reduced the arthritis score, inflammatory cells, erythrocyte sedimentation rate, and rheumatoid factor. Chrysin also reduced the mRNA levels of tumor necrosis factor, nuclear factor kappa-B, and toll-like recepter-2 and increased anti-inflammatory cytokines interleukin-4 and -10, as well as the hemoglobin levels. Using histopathology and microscopy, chrysin reduced the severity of arthritis in joints, infiltration of inflammatory cells, subcutaneous inflammation, cartilage erosion, bone erosion, and pannus formation. Chrysin showed comparable effects to piroxicam, which is used for the treatment of rheumatoid arthritis. The results showed that chrysin possesses anti-inflammatory and immunomodulatory effects that make it a potential drug for the treatment of arthritis.

## 1. Introduction

Worldwide, musculoskeletal diseases represent a major global health threat, affecting approximately 1.71 billion people; they are ranked as the leading cause of disability, measured by years lived with disability [1]. Lower- and middle-income countries are not immune to the burden of musculoskeletal diseases due to their struggling economies. In higher income countries, arthritis is associated with reduced workplace productivity; however, for residents of lower- and middle-income countries, arthritis imposes a potential additional burden by creating frequent visits to hospitals that subsequently worsen poverty because of days off work [2].

Rheumatoid arthritis (RA) is a chronic autoimmune inflammatory disorder that affects the joints symmetrically [3]. The synovial membrane, a soft tissue cover over the synovial cavity of the joint except for its cartilaginous parts, is the primary site of inflammation. The prevalence of RA ranges from 0.4% to 1.3% of the population, depending on age, sex, genetic predisposition, and environmental hazards [4]. The main symptoms of RA are joint pain, swelling, redness, and stiffness, particularly in the early morning or after prolonged sitting [5]. The synovial membrane is attacked by inflammatory cells during RA, causing rapid growth and thickening of the membrane with the development of pannus. Bone and cartilage erosion occur due to the release of enzymes from inflamed cells, leading to joint deformity with severe pain and loss of function. There is a 48% increased risk of cardiovascular diseases such as atherosclerosis and myocardial infarction and a 60% increased risk of premature death in patients with RA [6].

RA is a chronic inflammatory condition resulting from dysregulation of the immune response [7]. The cytokines, tumor necrosis factor (TNF) and interleukin (IL)-6 are central to the pathogenesis of RA. In addition, other cytokines such as IL-1β, IL-7, IL-17, IL-21, IL-23, granulocyte macrophage colony-stimulating factor, IL-18, IL-33, and IL-2 also play a role. The progression of RA appears to correlate with the serum levels of other chemokines, growth factors, and cytokines that regulate the inflammatory processes [8]. An imbalance between pro- and anti-inflammatory cytokines contributes to the induction of autoimmunity and chronic inflammation, which lead to joint destruction [9]. Several cytokine-mediated processes are involved in the development and regulation of inflammation in RA, including IL-4 mediated T-helper (Th) cell activation, IL-5 mediated macrophage polarization, B cell regulated IL-10 production, and IL-27 mediated inhibition of lymphoid follicle formation [10]. Toll-like receptors (TLRs) are pattern recognition receptors that identify multiple pathogen-associated molecular patterns and activate a variety of cell types [11]. TLRs play an important role in the pathogenesis of RA, and their expression is increased in RA, both in experimental models and in patients [12]. TLR-2 is highly expressed in the joint cavity, and TNF enhances the production of TLRs within the joint [13]. Although TNF plays a central role in osteoclast activation, it is also responsible for systemic symptoms of RA, such as fever, myalgia, and fatigue. TNF and TLRs induce the activity of many other transcription factors, particularly nuclear factor kappa B (NF-κB) [14]. NF-κB is thought to be involved in the transcription of more than 150 genes. NF-κB increases inflammation, leading to cartilage damage and bone erosion [15].

Currently available therapies for RA include non-steroidal anti-inflammatory drugs (NSAIDs), corticosteroids, disease-modifying anti-rheumatic drugs (DMARDs), and biologics such as anti-TNF agents, IL-6 receptor-inhibiting monoclonal antibodies, and selective JAK-STAT pathway inhibitors. These treatments slow disease progression by interrupting signals to the immune system to dampen the inflammatory processes, which result in joint damage [16]. Although NSAIDs were once considered first-line treatments for RA because they could relieve pain and inflammation, their use is associated with two major adverse events. First, despite their analgesic effect, NSAIDs do not address the underlying disease processes that drive the pathophysiology of RA. Second, the continuous use of NSAIDs potentiates the risk of serious side effects on the gastrointestinal and cardiovascular systems [17]. Piroxicam is an NSAID that is used to treat RA and has been widely used as a reference drug [18]. Piroxicam works similarly to other NSAIDs by inhibiting tissue cyclooxygenases (Cox-1 and -2) and reducing the production of prostaglandins, which are important mediators of pain and inflammation. Along with having analgesic properties, Piroxicam also possesses antipyretic and anti-inflammatory properties. With several million prescriptions written each year, Piroxicam was approved for use in the United States in 1982 and is still frequently used today. Similar to other NSAIDs, however, it might cause adverse effects such as headache, abdominal discomfort, dizziness, dyspepsia, and hypersensitivity reactions. Rare but serious adverse events from NSAIDs include increased risk for cardiovascular disease, gastrointestinal ulceration, and renal dysfunction [19].

Medicinal plants play a significant role in human health due to their extensive therapeutic uses in Ayurvedic, allopathic, and homeopathic medicines. Extracts from medicinal plants have made revolutionary changes in the field of medicine and greatly improved the pharmacological and pharmacokinetic properties of some drugs [20]. According to the World Health Organization (WHO), 80% of the world’s population relies on traditional medicine through the use of plants to treat diseases, and this dependence is increasing day by day due to their cost-effectiveness, safety, and high quality compared with synthetic drugs [21]. Different medicinal plants and their constituents possess many pharmacological properties such as analgesic, anti-pyretic, anti-inflammatory, anti-arthritic, anti-microbial, anti-proliferative, anti-cancer, anti-hemolytic, anti-depressant, and anti-oxidant properties [22].

Chrysin (5,7-dihydroxyflavone; C_15_H_10_O_4_) is a natural flavonoid that is obtained from passion flowers, propolis and honey wax, and honey [23]. Chrysin possesses a wide range of pharmacological activities, including anti-inflammatory [24], antibacterial [25], antioxidant [26], antiviral, immunomodulatory [27], antitumor [28], hypoglycemic, and other activities [29]. Multiple signaling pathways in cancer, including PI3K-Akt, Ras-Raf-MAPKs, Wnt-β-catenin, nuclear factor-kappaB (NF-κB), STAT, and Notch, are all modified by chrysin to limit cell proliferation, invasion, angiogenesis, and metastasis [30,31,32]. Chrysin can modify airway inflammation by regulating Th1/Th2 polarization through the inhibition of inducible nitric oxide synthase (iNOS) and NF-κB [33]. In addition, chrysin pretreatment inhibited extracellular signal-regulated kinase (ERK) phosphorylation, p38, and the release of inflammatory cytokines in airway inflammation [34]. The literature reveals that chrysin possesses anti-inflammatory effects by reducing the production of inflammatory mediators such as Cox-2, prostaglandin E2, and pro-inflammatory cytokines [35].

Animal models with adjuvant induced arthritis (AIA) are frequently used to analyze the efficacy of herbal products against RA. Complete Freund’s adjuvant (CFA)-induced arthritis shares many characteristics with human RA [36]. The current study aimed to assess chrysin’s anti-arthritic effects in CFA-induced joint damage in a pre-clinical model using rats.

## 2. Materials and Methods

### 2.1. Animals and Ethical Considerations

Male Wistar rats weighing 180–200 g and aged 6–8 weeks were used in this study. They were supplied by the Experimental Research Laboratory of the University of Health Sciences, Lahore. The animals were housed under standard laboratory conditions of 12-h day/night cycles at a temperature of 25 ± 3 °C with 60–70% humidity. The experimental protocol was approved by the Ethical Review Committee at the University of Health Sciences Lahore.

### 2.2. Experimental Design and Groups

Rats were divided into five groups: control, CFA arthritis, CFA arthritis plus chrysin 50 mg/kg, CFA arthritis plus chrysin 100 mg/kg, and CFA arthritis plus piroxicam 10 mg/kg (n = 10 per group). Chrysin (97%) and piroxicam (≥98%) were purchased from Sigma-Aldrich (St. Louis, MO, USA). The chemicals used in this investigation were of the highest analytical quality. RA was induced by injecting 0.2 mL of CFA (*Mycobacterium butyricum* dissolved in mineral oil) into the sub-plantar region of the left hind paw in all groups, except the control group [36]. Rats in the control group were injected with the same volume of normal saline. Animals in the low and high chrysin treatment groups received intraperitoneal injections of chrysin 50 mg/kg and 100 mg/kg daily, respectively [37,38], dissolved in 0.1% dimethyl sulfoxide (DMSO) from day 8 after injection of CFA to day 27. The piroxicam treatment group received piroxicam 10 mg/kg dissolved in 0.1% DMSO from day 8 after injection of CFA to day 27 [36].

### 2.3. Change in Body Weight

During the course of the experiment, the body weight of all rats was measured every week with a 0.1 g precision balance (METTLER TOLEDO, PB1501) and the change in body weight was assessed (g).

### 2.4. Arthritis Score Evaluation

The degree of arthritis was determined by the modified arthritis scoring methods of Mali et al., 2011 [39] and Zhu et al., 2014 [40]. The features of arthritis include swelling, redness, and erythema, which were determined by visual criteria on days 4, 8, 12, 16, 20, 24, and 28. A normal joint received a score of 0, while scores 1 to 4 were assigned based on the severity of swelling and redness.

### 2.5. Hematological and Biological Parameters

On the 28th day, blood samples (4 mL) were collected through direct cardiac puncture under light ether vapor anesthesia to analyze the hematological parameters. A hematology analyzer, Mindray BC-5150, was used to calculate the hemoglobin (Hb) and total leucocyte count (TLC). The erythrocyte sedimentation rate (ESR) was measured by the Wintrobe method [41]. Rheumatoid factor (RhF) levels were detected and semi-quantified through the agglutination method by a commercially available agglutination kit (Antec Diagnostics, UK).

### 2.6. Determination of mRNA Expression Levels of IL-4, IL-10, TNF, NF-κB, and TLR-2

Total RNA was extracted from 250 µL of blood by using TRIzol reagent (15596026; Thermo Fisher Scientific, Inc., Waltham, MA, USA). Optizen^TM^ NANOQ was used to perform a nano-drop test for the purity estimation and concentration of total RNA in each sample. Here, 1.5 µg of each RNA sample was reverse transcribed into cDNA by using the reverse transcription kit (K1621; Thermo Fisher Scientific, Inc., Waltham, MA, USA). The cDNA was amplified by polymerase chain reaction (PCR). The primer pairs used for RT-PCR are listed in Appendix A. PCR was performed using the following thermal cycling conditions: initial denaturation at 94 °C for 5 min, cyclic denaturation for 30 s at 94 °C, annealing for 30 s at 60 °C, and extension for 30 s at 72 °C (all steps, i.e., denaturation, annealing and extension were run consecutively for 40 cycles), and final extension for 10 min at 72 °C. The PCR products were resolved on agarose gel (2%) stained with ethidium bromide (111608; Sigma-Aldrich, St. Louis, MO, USA). In order to evaluate the differential expression pattern of TLR-2, NF-κB, TNF, IL-4, and IL-10, the relative densitometry values were calculated after normalization with GAPDH.

### 2.7. Histopathology of Ankle Joints

At the time of sacrifice, the ankle joints were preserved in a 10% formalin solution for fixation. Subsequently, the bone was decalcified using 10% boric acid in a 10% formalin solution and embedded in paraffin. Sections (5 μm) were cut onto glass histology slides, deparaffinized, and stained with hematoxylin and eosin (H&E) using routine procedures [39]. The slides were examined for the infiltration of inflammatory cells, synovial inflammation, subcutaneous inflammation, cartilage erosion, bone erosion, and pannus formation. A histopathologist (NN) blinded to experimental conditions scored the histopathologic changes as normal, minimal, mild, moderate, and severe changes (scores 0, l, 2, 3, and 4, respectively) [40].

### 2.8. Statistical Analysis

Data were analyzed by GraphPad Prism version 6 (GraphPad Software Inc., San Diego, CA, USA) and presented as mean ± standard deviation (SD). One-way and two-way analysis of variance (ANOVA) followed by *post hoc* Tukey’s test were applied to measure the significant mean difference among groups. A *p* value of ≤0.05 was considered statistically significant.

## 3. Results

### 3.1. Chrysin Normalized Body Weight in Arthritic Animals

Arthritis causes pain and swelling of the limb, which restricts the mobilization and feeding of animals. There was general weight loss in all CFA-treated animals (168 ± 17) on day 7 of arthritis induction compared with the control group (191 ± 35), but this loss in weight was not statistically significant. Treatment with chrysin (100 mg/kg) and piroxicam (10 mg/kg) improved the body weight, but this improvement was not significant compared with the arthritic group (*p*-value = 0.1335) (Figure 1a).

### 3.2. Chrysin Iinhibited the Extent of Arthritis

An “arthritis score” was devised from the extent of infiltration of inflammatory cells, synovial and subcutaneous inflammation, cartilage and bone erosion, and pannus formation in the joints. The results indicated a significantly higher arthritis score in CFA-treated rats compared with the untreated controls (*p*-value < 0.0001). After injecting CFA, early symptoms of inflammation were visible in the rats on day 8, and the arthritis score reached 3.938 ± 0.3 in 4 weeks. Arthritic rats showed an increase in thickness in their paws and ankles, which was evident by severe inflammation and edema. All of the groups treated with chrysin 50 mg/kg, chrysin 100 mg/kg, or piroxicam 10 mg/kg had a significantly-reduced arthritis score compared with CFA treatment (Figure 1b).

### 3.3. Chrysin Improved Hemoglobin Levels and Normalized Total Leukocyte Count in Arthritis

CFA caused a reduction in Hb levels in the arthritic group (11.63 ± 0.62) compared with the control group (14.76 ± 0.1). Chrysin 50 mg/kg (14.22 ± 0.15) and chrysin 100 mg/kg (13.73 ± 0.81) improved Hb levels after three weeks of treatment (*p*-value < 0.0001). These results were comparable with piroxicam 10 mg/kg (13.64 ± 0.47), as shown in Figure 2a. CFA caused a significant rise (*p*-value < 0.0001) in TLC in the arthritic group (17.27 ± 2.88) due to inflammation, compared with the control group (8.16 ± 1.66). Chrysin 50 mg/kg (4.01 ± 0.94) and chrysin 100 mg/kg (5.02 ± 1.29) improved inflammation and normalized TLC compared with the arthritic group. Piroxicam 10 mg/kg (11.02 ± 1.02) also caused a significant reduction in TLC (Figure 2b).

### 3.4. Chrysin Normalized the Erythrocyte Sedimentation Rate and Reduced Rheumatoid Factor

The results showed a significant rise (*p*-value = 0.0037) in ESR in the arthritic group (5.25 ± 0.95) compared with the control group (2.5 ± 1.2). There was a significant decrease in ESR in the chrysin 100 mg/kg treated (2.33 ± 1.2), chrysin 50 mg/kg treated (1.66 ± 0.81), and piroxicam treated (4.5 ± 0.57) groups compared with the arthritic group (Figure 3a). A significant rise (*p*-value < 0.0001) in RhF was noted in the arthritic group (33.3 ± 6) compared with the controls. After treatment with chrysin 100 mg/kg (3.3 ± 3), chrysin 50 mg/kg (5.3 ± 35), and piroxicam (12 ± 6), RhF was reduced significantly in all of the treated groups compared with the arthritic group (Figure 3b).

### 3.5. Chrysin Reduced mRNA Levels of TNF, NF-κB, and TLR-2

TNF, a pro-inflammatory cytokine that participates in the inflammatory process in the RA synovium and has systemic effects [42], was studied. As seen in Figure 4a, mRNA expression levels of TNF (1.79 ± 0.21) were increased significantly (*p*-value < 0.0001) in the arthritic group compared with the control animals. Treatment with chrysin 50 mg/kg (1.33 ± 0.15) and 100 mg/kg (1.25 ± 0.15) significantly modulated TNF levels. NF-κB is a key signaling molecule that plays a critical role in mediating articular bone erosion in RA [43]. The expression of NF-κB was the highest in the diseased group (1.68 ± 0.12), which was significantly different (*p*-value < 0.0001) from the control group. When compared with the diseased group, chrysin 50 mg/kg (1.18 ± 0.16), 100 mg/kg (1.26 ± 0.17), and piroxicam 10 mg/kg (1.27 ± 0.09) significantly reduced the NF-κB expression (Figure 4b). The expression of TLR-2 was significantly (*p*-value < 0.0001) upregulated in the arthritic group (2.74 ± 0.17) compared with the controls. After the treatment of 21 days with chrysin 50 mg/kg, chrysin 100 mg/kg, and piroxicam 10 mg/kg, the mRNA expression levels of TLR-2 were normalized in all of the treated groups (Figure 4c)

### 3.6. Chrysin Increased mRNA Levels of the Anti-Inflammatory Cytokines IL-4 and IL-10

The expression levels of anti-inflammatory cytokines IL-4 and IL-10 are shown in Figure 4d,e. Rats with arthritis showed a marked downregulation of IL-4 (0.405 ± 0.08) and IL-10 (0.309 ± 0.03), indicating that inflammation was successfully induced [44]. Treatment with chrysin 50 mg/kg (0.778 ± 0.08), chrysin 100 mg/kg (0.769 ± 0.12), and piroxicam 10mg/kg (0.795 ± 0.16) significantly upregulated (*p* ≤ 0.001) the levels of IL-4 mRNA compared with the arthritic group (*p*-value < 0.0001). Similarly, chrysin 50 mg/kg (0.605 ± 0.08), chrysin 100 mg/kg (0.496 ± 0.07), and piroxicam 10 mg/kg (0.622 ± 0.14) significantly upregulated the IL-10 mRNA levels compared with the arthritic group (*p*-value < 0.0001).

### 3.7. Chrysin Improved the Histopathological Parameters of Arthritis

Histopathology slides had been examined for infiltration of inflammatory cells, synovial inflammation, subcutaneous inflammation, cartilage erosion, bone erosion, and pannus formation and were scored as normal, minimal, mild, moderate, and severe changes (scores 0, l, 2, 3, and 4, respectively). There was no sign of inflammation upon histology of the ankle joint of the ipsilateral rat paw in the control group, but the arthritic group showed significant changes (*p* ≤ 0.001) in the severity of inflammation, including the extent of the inflammatory cell infiltrate, subcutaneous and synovial inflammation, along with obvious cartilage and bone erosion and pannus formation. Chrysin and piroxicam decreased the signs of inflammation, but the chrysin 100 mg/kg treated group showed better results compared with the chrysin 50 mg/kg and piroxicam treated groups, as shown in Figure 5 and Figure 6.

## 4. Discussion

Chrysin (5,7-dihydroxyflavone), a naturally occurring flavonoid, has various biological activities, including anti-inflammatory, anti-bacterial, anti-viral, anti-tumor properties, among others [45]. There is growing evidence that chrysin also exhibits various biological effects on the immune system. The effects of chrysin on the molecular mechanisms of the immune system have been identified [35], which prompted us to undertake this current study, in which we evaluated the immunomodulatory and anti-inflammatory effect of chrysin on CFA-induced arthritis.

CFA-induced arthritis in rats exhibits many similarities to human RA. The injection of CFA induces swelling of joints, which causes a rise in temperature and painful limb movements [46]. Paw swelling reaches its maximum in 3–5 days after CFA induction, as reported previously [47]. Chrysin, at both low and high doses, shows an improvement in arthritis scoring and reduces the gross morphology of arthritis in joints. Morphological features of inflammation, such as redness and swelling of the joint, are significantly reduced after chrysin treatment. Similar results are reported in the literature for Dashanga Ghana, an ayurvedic compound formulation, using the same arthritis model as the one used in the current investigation [48].

CFA causes joint inflammation and leads to an increase in inflammatory cells in the rodent RA model. In human RA, Hb decreases while there is an increase in ESR and RhF [49]. Hb levels fall as a result of inflammation, which causes an increase in the production of white blood cells. Arthritic patients mostly present with anemia. The results of this study show a reduction in hemoglobin levels in arthritic rats, which signifies that anemia may be due to the destruction of premature RBCs in the spleen, a reduction in the erythropoietin levels, and reduced iron loading in the synovial tissue of the reticuloendothelial system [50]. Bone marrow shifts from erythropoiesis to leukocytopoiesis. ESR increases due to a rise in inflammatory cells in the blood [51]. RhF is an important parameter of RA. In the current investigation, increased RhF and ESR were identified with CFA-induced arthritis, and there was a rise in RhF after CFA induction, which causes arthritis [52].

The immune system is stimulated by CFA, with a complex set of antigenic signals to the innate compartment of the immune system. The outcome is altered leukocyte proliferation and differentiation, and the release of pro-inflammatory cytokines [53]. The presence of mycobacteria in the CFA stimulates macrophages and dendritic cells by inducing TNF and IL-10. TLR-2 contributes to the recognition of gram-positive bacteria components and lipopolysaccharides [54], while TLR-4 is involved in innate immunity. TLRs cause the activation of NF-κB, leading to the production of IL-1, IL-6, IL-8, and TNF [55]. TLR-2 has a vital role in the induction of apoptosis [56]. All chrysin- or piroxicam-treated groups showed a reduction in TNF, NF-κB, and TLR-2 levels after 21 days, but the results of high dose chrysin were better than those of low dose chrysin and piroxicam. Previous studies conducted on RA support these results [57].

The pathogenesis of RA involves multi-faceted immune regulation, and Th cell subsets such as Th1, Th2, Treg, Th17, and Tfh cells and their secreted cytokines are involved in immune regulation during the course of RA, playing a role in promoting inflammation or immune protection. In addition, there are many immunomodulatory agents that promote or inhibit the immunomodulatory role of Th cells in RA to alleviate the progression of RA. RA is a Th1 mediated disorder that activates Th1 cells including IL-1, TNF-α, and INF-ℽ, while Th2 opposes this process to maintain balance [40]. IL-4 and IL-10 are the main cytokines that regulate the functions of Th2 and provide beneficial effects against autoimmune disorders [58]. Our study also demonstrates a reduction in IL-4 and IL-10 levels due to CFA induction, while chrysin and piroxicam act as immunomodulatory agents and increase the levels of anti-inflammatory cytokines [32].

The histopathological data also revealed that chrysin not only reduces inflammation, but also prevents the cartilage and bones from erosion, which also favors the data obtained from other biochemical parameters. Infiltration of inflammatory cells, subcutaneous inflammation, and synovial inflammation are the characteristics of arthritis [38]. Treatment with chrysin as well as piroxicam reduces these inflammatory parameters significantly, but chrysin 100 mg/kg decreases the inflammation most effectively. The hallmarks of arthritis, i.e., cartilage erosion, bone erosion, and pannus formation, are modified by chrysin treatment. These changes are prominent in the arthritic group. Chrysin not only reduces inflammation, but also prevents the cartilage and bones from erosion. These results support the anti-rheumatoid effects of chrysin. Bone and cartilage erosion is mainly influenced by oxidative damage, as it reduces bone formation by minimizing the differentiation and viability of osteoblasts and by enhancing reactive oxygen species mediated by osteoclasts activation. IL-1β, IL-6, and TNF are the main stimulants of bone erosion and are the main inhibitors of bone formation [59,60]. A fluctuation in IL-6 levels determines the severity of inflammatory response under stress, such as tissue injury, whereas IL-1β and TNF act synergistically to enhance other cytokines. Increasing pro-inflammatory cytokines also contribute to the increased number of osteoclasts in bone tissue, thus promoting the imbalance of bone metabolic coupling [61].

Excessive inflammation leads to bone damage and loss of mobility [18,62]. However, chrysin can attenuate these changes. Previous literature and our results, as well, suggest that chrysin and piroxicam may reduce bone damage by attenuating pro-inflammatory cytokine production, upregulating anti-inflammatory cytokine levels, and by improving oxidative status, which ultimately promote the growth, differentiation, and proliferation of rat osteoblasts. Furthermore, it seems that chrysin may reduce the incidence of osteoporosis by ameliorating the interactions between the bone and immune system. One of the rarely considered benefits of the flavonoids, which include chrysin, is their effect on the microbiome [63]. As a prebiotic, chrysin may promote calcium and phosphorus absorption through the intestine. Prebiotics therefore impact bone health perhaps by promoting the release of bone modulation factors and suppressing bone resorption [64].

## 5. Conclusion

This study demonstrates that chrysin has anti-inflammatory and immunomodulatory characteristics and RA disease-modifying properties. Its effects are comparable with those of the NSAID, piroxicam. Chrysin may be a better treatment option for chronic autoimmune diseases than corticosteroid injections in the future; however, further studies should be conducted to compare the effects of chrysin with those of dexamethasone or methotrexate in order to evaluate its RA disease modifying potential against these established DMARDs.

## Figures and Tables

**Figure 1 pharmaceutics-15-01225-f001:**
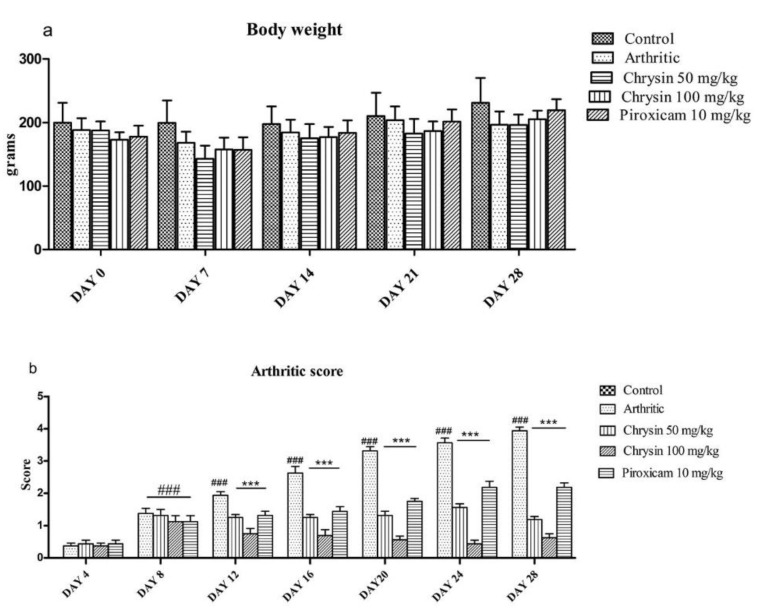
Comparison of body weight (**a**) and arthritis score (**b**) in all experimental groups (n = 10 per group). ### indicates *p*-value ≤ 0.001 and shows significant difference from the untreated controls. *** indicates *p*-value ≤ 0.001 and shows a significant difference from the CFA arthritic group.

**Figure 2 pharmaceutics-15-01225-f002:**
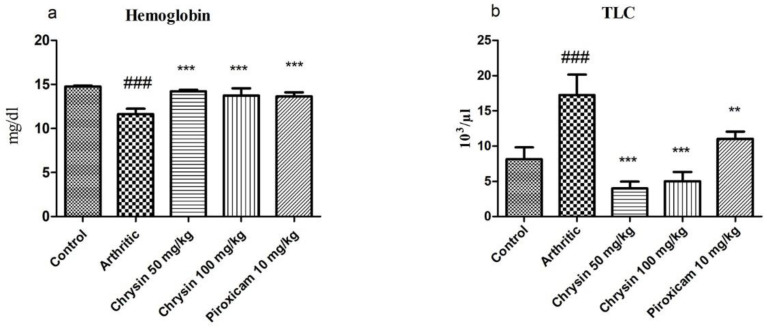
Comparison of hemoglobin (**a**) and total leucocyte count (TLC) (**b**) in all experimental groups (n = 10 per group). ### indicates *p*-value ≤ 0.001 and shows significant difference from the untreated controls. ** indicates *p*-value ≤ 0.01 and *** indicates *p*-value ≤ 0.001. These show a significant difference from the CFA arthritic group.

**Figure 3 pharmaceutics-15-01225-f003:**
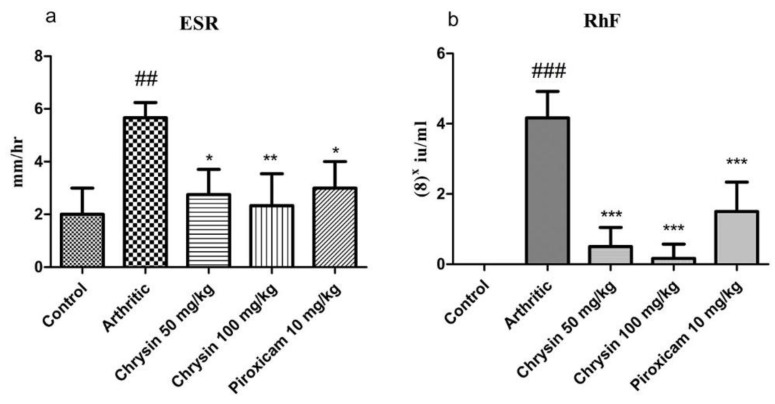
Comparison of erythrocyte sedimentation rate (ESR) (**a**) and rheumatoid factor (**b**) in all experimental groups (n = 10 per group). ## indicates *p*-value ≤ 0.01, and ### indicates *p*-value ≤ 0.001, showing a significant difference from control. * indicates *p*-value ≤ 0.05, ** indicates *p*-value ≤ 0.01 and *** indicates *p*-value ≤0.001 and they indicate a significant difference from the CFA arthritic group.

**Figure 4 pharmaceutics-15-01225-f004:**
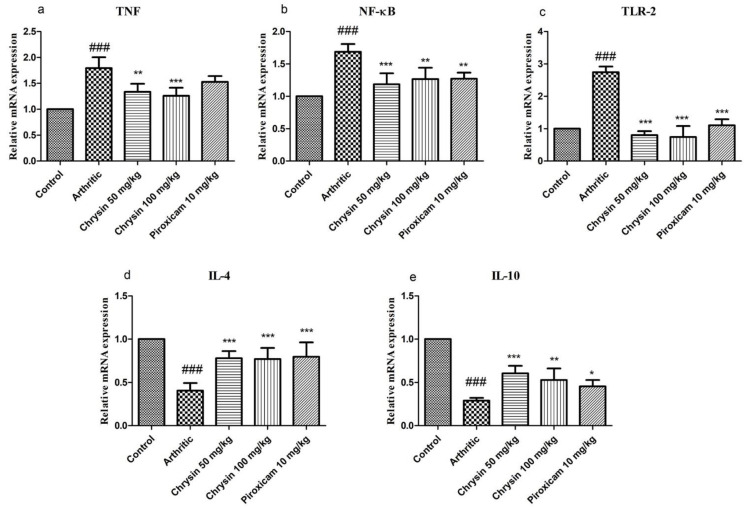
Comparison of relative mRNA expression level of TNF (**a**), NF-κB (**b**), TLR-2 (**c**), IL-4 (**d**), and IL-10 (**e**) in all of the experimental groups (n = 10 per group). ## indicates *p*-value ≤ 0.01, ### indicates *p*-value ≤ 0.001, and they show significant differences from the control group. * indicates *p*-value ≤ 0.05, ** indicates *p*-value ≤ 0.01, and *** indicates *p*-value ≤ 0.001, and they show a significant difference from the CFA arthritic group.

**Figure 5 pharmaceutics-15-01225-f005:**
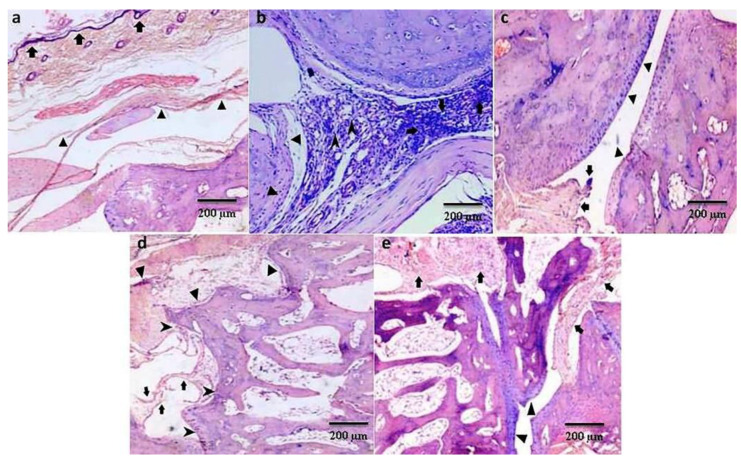
Histological images (H&E 
stain, 100X) of examples of ankle joints of different groups including the control 
group (**a**), arthritic group (**b**), chrysin 50 mg/kg (**c**), 
chrysin 100 mg/kg (**d**), and piroxicam 10 mg/kg groups (**e**) (n = 10 
per group). The control group shows no inflammation. Using the extent of 
inflammatory cell infiltration as an indicator of effect, the CFA arthritic 
group shows severe inflammation/extensive inflammatory cell infiltration; while 
(**c**–**e**) show an improvement in the extent of inflammatory cell 
infiltration after treatment with chrysin (50 mg/kg or 100 mg/kg, respectively) 
or piroxicam in CFA arthritic rats. (**a**): 
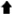
 shows 
stratified squamous keratinized epithelium. (**b**): ▲ Shows chronic 
granulomatous inflammation, 
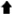
 Shows pannus 
formation, 
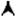
 Shows peri-articular 
inflammation. (**c**): ▲ Showing synovial hyperplasia, 
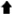
 Shows articular 
cartilage with no inflammation. (**d**): 
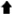
 = mild 
synovial hyperplasia, ▲ = mild peri articular inflammation, 
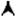
 = intact articular 
cartilage. (**e**): 
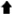
 shows mild synovial 
inflammation and hyperplasia, ▲ shows intact articular cartilage.

**Figure 6 pharmaceutics-15-01225-f006:**
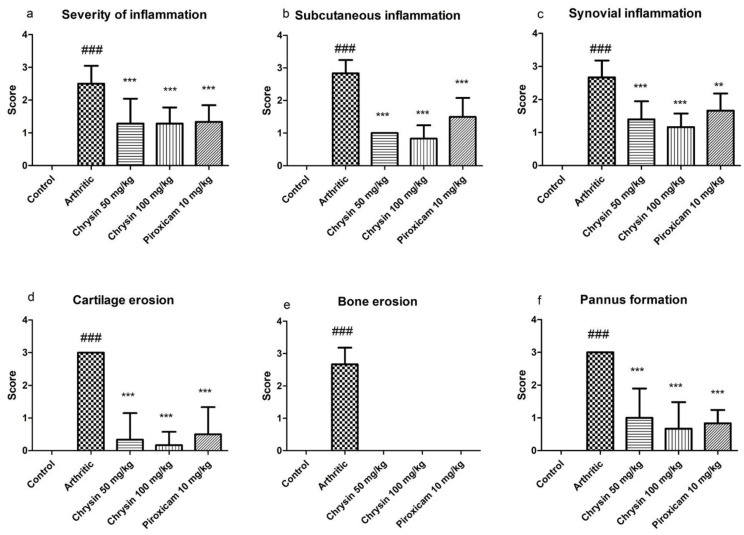
Comparison of histological scoring while assessing severity of inflammation (**a**), subcutaneous inflammation (**b**), synovial inflammation (**c**), cartilage erosion (**d**), bone erosion (**e**), and pannus formation (**f**) in rats of all experimental groups (n = 10 per group). ## indicates *p*-value ≤ 0.01, ### indicates *p*-value ≤ 0.001 showing significant difference from control. ** indicates *p*-value ≤ 0.01 and *** indicates *p*-value ≤ 0.001 showing a significant difference from the CFA arthritic group.

## Data Availability

The data used to support the findings of this study are available from the corresponding author (MS) upon request.

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
