# Peer review of "Chrysin Is Immunomodulatory and Anti-Inflammatory against Complete Freund’s Adjuvant-Induced Arthritis in a Pre-Clinical Rodent Model"

_pharmaceutics, 2023, doi:10.3390/pharmaceutics15041225_

Round 1
Reviewer 1 Report
Faheem et al. describe the effect of chrysin on rheumatoid arthritis. The manuscript present evidence that Chrysin treatment reduces arthritis symptoms. Nevertheless, there a several points to be addressed prior publication.
The introduction summarize the most important facts about rheumatoid arthritis (RA), but the paragraph describing the effects of IL-4, IL-5 and IL-10 has to be reorganized. In the current version it appears that these cytokines are main disease drivers. That is not correct, they are proposed to have anti-inflammatory or immune regulatory effects. Cytokines critically involved in disease pathogenesis are TNF-α, IL-6 or IL-1β.
Is TLR2 also important for disease progression in humans or is it a rodent specific effect?
In the paragraph about current available therapies for RA, the authors did not mention anti-IL-6 antibodies and JAK-STAT inhibitors.
Why the authors use the NSAID Piroxicam as reference drug and not an established disease-modifying drug, such as methotrexate? NSAIDs do not address disease-driving processes as stated correctly in the introduction. Therefore, the positive effects of Piroxicam on arthrits score presented in figure 1b are a little bit surprising. Moreover, the authors do not discuss this phenomenon at all.
As the authors do not perform quantitative real time PCR, they have to present gel photos of the PCR products to prove their results.
The bars in figure 1b are hard to distinguish. Maybe the authors could modify the fill pattern of the bars.
According to figure 3a, the effect of Piroxicam on ESR is significant. The authors claim the opposite in the text.
Which staining method was used for histology presented in figure 5? The information should be added to the figure legend. In addition, scale bars in the photographs are missing. As well as more detailed description, which parameters in the slides indicate less inflammation. A reduced number of infiltrating cells?
I would suggest to add in all figure legends the number of animals analyzed.
In the discussion, the paragraph about bone resorption is too detailed and no obvious link to their experimental results exist. The same is true for the microbiome paragraph.
Author Response
the comments are attached below

Reviewer 2 Report
I found that this manuscript is totally relevant for the field and brings interesting results that deserve consideration. However, certain points need to be revised before the manuscript could be considered for publication.
· Please, provide the source of experimental animals.
· Please, provide the amount of blood used for RNA extraction.
· In materials and methods section, there are many primers involved in the real-time PCR-PCR test. It is recommended to present them in the form of a table provided with their size, GenBank Accession Numbers or references.
· In the real-time PCR-PCR test, please mention the method used to determine the relative expression levels of each gene.
· Please give specific category numbers of all chemicals and kits used in this study so people can cite and repeat the work in the future if wanted.
· All p in the manuscript should be in italics and capital.
· P-value should be P<0.05 not P< 0.05.
· Did all the data meet the ANOVA assumptions? The realization of the ANOVA for independent groups requires the analysis of the assumptions of independence of observations, normality and homogeneity of variances. It is unclear in the manuscript whether these assumptions were met.
· In "results", authors should provide complete statistical information: exact p-value, F-value, degrees of freedom, etc.).
· Figure legends (1-4&6) should be rewritten with mention the value of n.
· Legend of Fig 5, the authors should mention the magnification power and name of stain of tissue used in this study.
Author Response
the comments are attached below

Round 2
Reviewer 1 Report
The authors answered most of my questions adequately. Nevertheless, some points have to be addressed prior publication.
The authors used the NSAID Piroxicam as a positive control. By their analgesic and anti-inflammatory effects NSAIDs ameliorates pain and inflammation in RA patients. In their experiments the authors demonstrate effects of Piroxicam on bone erosion and pannus formation (figure 5 and 6). These implies an effect of NSAID on disease driving processes, which in human RA patients is not observed.
How the authors explain this finding? Chrysin effects are almost comparable to that of Piroxicam. Is Chrysin therefore a “new” NSAID like drug? What are the advantages of Chrysin compared to Piroxicam? Why the authors believe, that Chrysin might belong rather to the group of disease-modifying drugs? The authors have to discuss this point.
The gel photos of the PCR products should be part of the main manuscript.
Line 71: “Several cytokine like the ILs.” Please rephrase this sentence.
Line 72-75: Please check grammar.
Line 85: “TNF and TLRs induce the activity of many other cytokines, particularly nuclear factor kappa B (NF-κB)”. NF-kB is a transcription factor and not a cytokine.
Figure 1B: In the figure legend, the symbol # is not explained.
Line 375: “FCA” Please change in CFA.
Author Response
Response to reviewer is attached in a file bellow

Reviewer 2 Report
The revised manuscript is greatly improved, l recommend the paper for publication.
Author Response
.